# The Changbai Alpine Shrub Tundra Will Be Replaced by Herbaceous Tundra under Global Climate Change

**DOI:** 10.3390/plants8100370

**Published:** 2019-09-25

**Authors:** Yinghua Jin, Jiawei Xu, Hongshi He, Mai-He Li, Yan Tao, Yingjie Zhang, Rui Hu, Xiang Gao, Yunyu Bai, Huiyun Wang, Yingying Han

**Affiliations:** 1Key Laboratory of Geographical Processes and Ecological Security in Changbai Mountains, Ministry of Education, School of Geographical Sciences, Northeast Normal University, Changchun 130024, China; jinyh796@nenu.edu.cn (Y.J.); heh@missouri.edu (H.H.); maihe.li@wsl.ch (M.-H.L.); taoy431@nenu.edu.cn (Y.T.); zhangyj486@nenu.edu.cn (Y.Z.); hur470@nenu.edu.cn (R.H.); gaox170@nenu.edu.cn (X.G.); baiyy430@nenu.edu.cn (Y.B.); wangh0744@nenu.edu.cn (H.W.); hanyy104@nenu.edu.cn (Y.H.); 2School of Natural Resources, University of Missouri, Columbia, MO 65211, USA; 3Swiss Federal Institute for Forest, Snow and Landscape Research WSL, CH-8903 Birmensdorf, Switzerland

**Keywords:** alpine tundra, herbaceous plant expansion, dominant species, spatial distribution, altitudinal and topographic factors, Changbai Mountains

## Abstract

Significant replacement of shrub species by herbaceous species has been observed in the Changbai alpine tundra zone, China, since the 1990s. This study used plot surveys to analyze variations in the spatial distribution of dominant plants and to ascertain the changing mechanisms of dominant species in the alpine tundra zone. We found that the two previously dominant shrubs, *Rhododendron chrysanthum* and *Vaccinium uliginosum*, differed markedly in their distribution characteristics. The former had the highest abundance and the lowest coefficient of variation, skewness, and kurtosis, and the latter showed the opposite results, while the six herb species invaded had intermediate values. *R. chrysanthum* still had a relatively uniform distribution, while the herbaceous species and *V. uliginosum* had a patch distribution deviating from the normal distribution in the tundra zone. Micro-topography and slope grade had stronger effects on the spatial distribution of the eight plant species than elevation. Herbs tended to easily replace the shrubs on a semi-sunny slope aspect, steep slope, and depression. Overall, the dominance of dwarf shrubs declined, while the herbaceous species have encroached and expanded on the alpine tundra zone and have become co-dominant plant species. Our results suggest that various micro-topographic factors associated with variations in climatic and edaphic conditions determine the spatial distribution of plants in the alpine tundra zone. Future climate warming may cause decreased snow thickness, increased growing season length, and drought stress, which may further promote replacement of the shrubs by herbs, which shows retrogressive vegetation successions in the Changbai alpine tundra zone. Further studies need to focus on the physio-ecological mechanisms underlying the vegetation change and species replacement in the alpine tundra area under global climate change.

## 1. Introduction

Alpine tundra environments are characterized by the cold climate, short growing season, intense sunlight, large differences in day and night temperatures, and high precipitation [1]. Alpine tundra soils tend to be poorly developed and thin. The alpine tundra is an extremely fragile habitat. The alpine tundra is vulnerable to outside interference and forms an ecologically sensitive zone that responds significantly to changing environmental conditions such as global warming, acid deposition, and atmospheric nutrient inputs [2,3,4].

Significant changes in climate and vegetation have occurred in alpine and sub-alpine areas worldwide [5,6,7,8,9,10]. Over the past several decades, many species, especially herbs, from lower elevations have extended their ranges into higher elevations such as alpine and subalpine areas [11,12,13,14,15,16]. In particular, this upward migration trend is accelerating under currently rapid global warming [17,18,19,20,21,22].

The tundra of the Changbai Mountains is one of the most typical mountain tundras in Asia [23]. The local climate has undergone a significant change over the past 30 years [24,25,26,27], including increased temperature during the growing season, increased accumulated temperature, longer growing season [28], increased precipitation and increased precipitation intensity [27,29], decreased snowpack, earlier snowmelt, and a shortened snow period [30].

The frigid climate and thin mountain soil support polar or alpine plant species that account for about 80% of the total plant species in the Changbai alpine tundra. Qian and Zhang (1980) [31], Huang and Li (1984) [23], and Qian (1990) [32] investigated the community structure and identified the dominant species in the tundra and results showed that the dominant shrub species were *Dryas octopetala*, *Phyllodocecaerulea*, *Rhododendron chrysanthum*, *Rhododendron confertissimum*, *Rhododendron redowskianum*, *Vaccinium uliginosum*, and *Vaccinium vitis-idaea*. The dominant plants are mainly shrubs (*R. chrysanthum* and *V. uliginosum*), and the community has a shrub and a moss-lichen layer, but lacks an herbaceous layer [31,32]. Therefore, the alpine tundra of the Changbai Mountains is considered as a shrub tundra. Herbaceous plants such as *Calamagrostis angustifolia* form the lower elevation ‘*Betula ermanii* zone’ have invaded the alpine tundra zone on the Changbai Mountains since the 1990s [33]. *C. angustifolia* was not recorded by Qian’s surveys [31], but *C. angustifolia* individuals have been found in a few plots surveyed by Huang (1984) [23], and is now widespread and distributed in the tundra [33]. Zong et al. (2013) [33] revealed, based on spectral and image analysis, that the incursion of *C. angustifolia* from lower elevations to higher elevations started in the 1980s, and has successfully invaded the alpine tundra landscape since the 21st century. Currently, the *C. angustifolia* patches at lower elevations under the tundra are interconnected, and form relatively large patches after years of expansion. At higher elevations in the tundra landscape, more herbaceous species such as *C. angustifolia* encroached on the tundra zone [34,35,36]. Jin et al. (2016) [36] suggest that herbaceous species have now been forming patches on the alpine tundra zone and becoming a co-dominant plant species with shrubs that have correspondingly been becoming fragmented and patchy. Therefore, the tundra seems to be transforming into alpine meadows [36,37].

As mentioned above, previous descriptive studies on the tundra vegetation of the Changbai Mountains indicated that the tundra vegetation has changed significantly over the last 30 years. However, there is a lack of quantitative study on the composition and spatial distribution of tundra vegetation. Several surveys in the tundra were conducted, but the exact locations investigated were not well documented [23,31], and, thus, a comparable re-investigation is impossible. In the early 1980s, the tundra community is nearly purely composed of shrubs [23,31]. Recently, some studies found that eight plant species dominated the tundra, including two shrub species (*R. chrysanthum* and *V. uliginosum*) and six herb species (*C. angustifolia, Geranium baishanense, Ligularia jamesii, Sanguisorba parviflora, Sanguisorba stipulata*, and *Saussurea tomentosa*) [38,39]. These herb species previously either occurred in the mountain birch forests at a lower elevation or were occasionally observed in the tundra [39]. The distribution patterns of these eight dominant species belong to the aggregated distribution, but the aggregate intensity varies among species. The aggregate intensity of *R. chrysanthum* is the weakest, while the aggregate intensity of herbs is usually stronger. Moreover, the associations among six dominant herbs are different and their associations are low [37]. This phenomenon indicates that there may be differences in habitat selection among dominant herbs and, thus, they may occupy various spatial locations. However, the variation characteristics and differences in the spatial distribution of dominant plants in that tundra remain unknown.

The main factors affecting the composition of alpine plant communities are altitude, topography, and those-associated soil development [40,41]. Altitude is the main variable affecting floristic diversity in the communities as a whole, but an individual species existing in a community is more influenced by topography [42]. From wind-exposed ridges to small micro-sites, the effects of topography were found to create various ecological conditions with different temperature, snow accumulation, and water availability, as well as soil development and soil nutrient supply [43,44,45]. Various combinations of these factors affect species distribution [46], spatial structure [47], and plant responses to climate change [18,48].

To understand the factors affecting distribution and immigration of plants, and, thus, to predict the future vegetation composition of the Changbai Mountain tundra, we examined the characteristics and differences in the spatial distribution of dominant plants in that tundra. We used the descriptive statistics of all dominant species to study the variation characteristics of dominant species in the community and the change process of the tundra community. We performed one-way analysis of variance (ANOVA) and generalized linear models (GLMs) to test the differences in the spatial distribution of the dominant species in elevation, slope aspect, slope grade, and micro-topography. We conducted redundancy analysis (RDA) of the dominant species and environmental factors to find the main factors determining the spatial distribution of the dominant species. After that, we analyzed the mechanisms underlying changes in the tundra vegetation of the Changbai Mountains. We aimed to test the hypothesis that topography rather than other environmental factors determine the herbaceous vegetation expansion in the tundra of the Changbai Mountains. Our results of vegetation changes, process, and mechanisms of the Changbai alpine tundra can provide a better understanding of how the tundra vegetation will respond to global climate change.

## 2. Results

### 2.1. Statistical Characteristics of the Dominant Species

Among the eight dominant species, *R. chrysanthum* had the highest abundance and the smallest coefficient of variation, skewness, and kurtosis (Table 1 and Table 2). *R. chrysanthum* had a relatively uniform and wide distribution, which was close to a normal distribution in the tundra zone. By contrast, *V. uliginosum* had the lowest abundance but the largest coefficient of variation, skewness, and kurtosis among the eight species (Table 2). *V. uliginosum* had the largest positive skewness. Its distribution is positively skewed with a mean value to the right of the peak value. *V. uliginosum* had larger abundance only in a few plots and had the highest positive kurtosis (Table 2).

The dominant herbaceous plants had an intermediate abundance and coefficient of variation, skewness, and kurtosis (Table 2). The plant distribution characteristics of herbs were between those of *R. chrysanthum* and *V. uliginosum*, and the statistical characteristics of the six herb species varied with species. The abundance of *S. parviflora* was the highest among the six herb species, and its coefficient of variation, skewness, and kurtosis were the lowest (Table 2). *L. jamesii* had an intermediate abundance but the highest coefficients of variation, skewness, and kurtosis are among the six herbs (Table 2).

### 2.2. Spatial Distribution of the Dominant Species

#### 2.2.1. Altitudinal Factor

Except for *R. chrysanthum* and *S. parviflora*, elevation significantly affected the abundance of the other six species (Table 3). Differences existed in the distribution of the two dominant shrub species along the elevational gradient in the strip. The abundance of *R. chrysanthum* was the highest, while the abundance of *V. uliginosum* was the lowest along the elevational gradient among the eight dominant species (Figure 1). The abundance of *R. chrysanthum* did not differ with elevation (*p* > 0.05), whereas the abundance of *V. uliginosum* was significantly different along the elevational gradient (*p* ≤ 0.01) (Table 3 and Figure 1).

The abundance of the six dominant herb species were intermediate between the two dominant shrub species along the elevational gradient (Figure 1). The abundance of the six herb species varied significantly among elevations (Table 3, Figure 1). *L. jamesii* and *G. baishanense* decreased their abundance with increasing elevation (Figure 1).

The shrub/herb ratio was not affected by elevation (*p* > 0.05), but a significant difference in the Shannon-Wiener index was observed among the elevational gradient (*p* ≤ 0.05) (Table 3).

#### 2.2.2. Slope Aspects

Slope aspects significantly affected the abundance of the two shrubs (Table 3). The abundance *R. chrysanthum* were large on the north-facing and south-facing slope. The abundance of *V. uliginosum* were low on all slope aspects, especially, rarely seen on the southeast-facing slope, southwest-facing and east-facing slopes (Figure 2). No significant difference in the abundance of *R. chrysanthum* was observed between those on the north-facing and south-facing slope (*p* > 0.05), but there were significant differences in those on west-facing, southwest-facing, southeast-facing, and east-facing slopes (*p* ≤ 0.05). The abundance of *V. uliginosum* on the north-facing slope differed significantly with those on other slopes (*p* ≤ 0.05) (Table 3 and Figure 2).

The six herb species appeared with low abundance on all slope aspects. In particular, it was the least in the north-facing slope. Among them, the abundance of *C. angustifolia* and *S. stipulate* are high in the southeast-facing and east-facing slopes (Figure 2). The abundance of the three dominant herb species (e.g., *S. parviflora*, *C. angustifolia*, and *S. tomentosa)* were not significantly different among slope aspects (*p* > 0.05), but the abundance of the other three herb species were significantly different among slope aspects (*p* ≤ 0.05) (Table 3 and Figure 2).

The shrub/herb ratio and the Shannon-Wiener index were not significantly different among slope aspects (*p* > 0.05) (Table 3).

#### 2.2.3. Slope Grade

Except for *S. tomentosa*, slope grade significantly affected the distribution of seven dominant species (*p* ≤ 0.05) (Table 3 and Figure 3). Among the eight species, *R. chrysanthum* had the largest abundance across all slope grades (Figure 3) and were mainly distributed on slopes below 35°. The distribution of *V. uliginosum* concentrated on slopes below 5°. However, the majority of the herbs concentrated on steep slopes (above 35°) (Figure 3).

The shrub/herb ratio was significantly different (*p* ≤ 0.05), but the Shannon-Wiener index was not significantly different among slope grades (*p* > 0.05) (Table 3).

#### 2.2.4. Micro-Topography

Among the eight species, the abundance of *R. chrysanthum* and *S. parviflora* were not affected by the micro-topography (*p* > 0.05) (Table 3). *R. chrysanthum* had the largest abundance in different micro-topographies. *V. uliginosum* was high in the convex (Figure 4). The abundance of the majority of the six herbs were larger in depression and transitional zones (Figure 4).

The micro-topography significantly influenced the shrub/ herb ratio (*p* ≤ 0.05), but it did not affect the Shannon-Wiener index (*p* > 0.05) (Table 3).

### 2.3. Correlation Analysis of Dominant Species and Altitudinal and Topographic Factors

The RDA ordination results of these species and the four factors showed that the eigenvalues of the first two ordination axes were 0.129 and 0.021, respectively, which account for 88.1% of the cumulative variance in the species–environment relation (Table 4).

The four factors affected the distribution of the eight species in different contents, with the strongest effects of micro-topographies (Table 4 and Figure 5). Among the four factors, micro-topography was most highly correlated with the first ordination axis, followed by slope grade (Table 4 and Figure 5). Slope grade was mostly correlated with the second ordination axis, followed by micro-topography (Table 4 and Figure 5).

As a whole, shrub/herb ratio was strongly correlated with micro-topography and slope grade (Table 5 and Table 6). *R. chrysanthum* was mostly correlated with the slope aspect, whereas *V. uliginosum* and the majority of the herbs had a strong correlation with micro-topography (Table 5 and Figure 5).

## 3. Discussion

### 3.1. Expansion of Herbaceous Plants

In this study, we used plot surveys to analyze the distribution characteristics of eight dominant plants in the tundra zone of the Changbai Mountains. The tundra vegetation of the Changbai Mountains is changing. Herbaceous species have encroached and expanded on the alpine shrub tundra vegetation. The abundance of herbaceous plants varied with the elevation and micro-topography, which implies the effects of micro-site associated with climatic and edaphic conditions on plant distribution. Tundra ecosystems are commonly regarded as being highly sensitive to global climate change. Observed changes in plant communities in both Arctic and alpine tundra environments have been associated with recent climate warming [49]. Previous studies suggest that individual species in a community respond differently rather than cohesively to directional climate change [5]. Our previous results showed that *C. angustifolia* has significantly invaded the shrub tundra zone of the Changbai Mountains during the last 30 years [33,38]. By the 1980s, Changbai alpine tundra was defined as shrub tundra dominated by *R. chrysanthum* and *V. uliginosum* with normal distribution, and there was only sparse herbaceous species in the community [31]. The present study showed that the dominance of dwarf shrubs has declined. Though *R. chrysanthum* was still the most important dominant species, the dominance of *V. uliginosum* declined seriously. The dominance of six herbaceous species has increased significantly and were between those of *R. chrysanthum* and *V. uliginosum*, which indicates that the herbaceous plant species have encroached on tundra and predominated in the tundra. The shrub community in the tundra is undergoing a major change.

The descriptive statistics characteristics (mean, coefficient of variation, skewness, and kurtosis) of abundance for the two shrub dominant species indicated that *V. uliginosum* was more sensitive than *R. chrysanthum*. *R. chrysanthum* has a strong resistance to herbaceous plant expansion and it still had a relatively uniform and wide distribution in the tundra zone. The abundance of *V. uliginosum* showed the largest positive skewness and kurtosis, which indicates that *V. uliginosum* had a patch distribution in a particular habitat.

Statistical characteristics of the six herb species suggested that their invasion was not synchronous. *S. parviflora* invaded the tundra earlier and, thus, it has time to establish with a stable distribution, which indicates a late invasion stage, and it has become a primary dominant species among the six herb species. Compared to *S. parviflora*, the distribution of *C. angustifolia* reflected a mid-stage of invasion, and *S. tomentosa* and *G. baishanense* were at an early invasion stage, which are starting to invade the tundra.

Alpine tundra ecosystems are controlled by a low temperature. The lower temperature with increasing altitude accounts for more than 80% of the variation among summits to explain the community species composition [46]. Current global warming directly affect temperature in the tundra zone, and indirectly change the precipitation and snow cover, which leads to changes in species’ composition. Species with similar distributions respond similarly to large and local ecological gradients [50]. Biogeographic deconstruction has been used to assess community patterns of species indifferent habitats and their ecological requirements [51,52]. Compared with previous analysis at the community level in the Changbai alpine tundra, our analysis at the species level provided a better understanding of the influence of altitude and micro-topography on the changes in the distribution of shrub and herbaceous plants, and shift of the tundra species’ composition under climate change. For example, six herbaceous species have already occupied lower elevations, a semi-sunny slope aspect, steep slope, and depression in that tundra, which shifted the shrub tundra to the shrub-herb tundra. Our results showed that the Shannon-Wiener index was mainly affected by the altitude, whereas the shrub/herb ratio and six herbaceous dominant species were significantly affected by micro-topography, slope grade, and slope aspects. These findings are consistent with earlier studies [42].

### 3.2. Implication for Possible Causes of Changes in the Tundra Vegetation

Our results indicated that herbaceous species gradually migrated from lower elevations to tundra at high elevations. In the cold conditions of the tundra of the Changbai Mountains, shrubs have a competitive advantage over herbs [23]. With global warming, the growing season temperature, and the length of the growing season, have increased and are continuously increasing in the tundra of Changbai Mountains [27]. These factors allow historically low elevation species (herbs) to survive at higher elevations. Our results are consistent with those of previous studies where air warming resulted in herbaceous plant invasion into alpine and arctic tundras [21,42,53,54].

Our results showed that herbs typically existed on semi-sunny slopes, steep slopes, and depression. It means that micro-topography and slope grade play a greater role in determining herbs invasion, which supported previous results [22,42,55]. By the 1980s, shrubs covered almost all areas of the tundra, except for patches covered by permanent snow on the Changbai Mountains [23]. After 30 years of vegetation change, shrubs have been replaced by herbs in a considerate part of the tundra. The reason might be effects of tundra climate change associated with air warming, increased precipitation, and decreased snow cover on vegetation [33]. For instance, shortened snowfall season and reduced snow cover [33], less snow, and thinner winter snow cover weaken the protective effects on survival of *R. chrysanthum* [46,56].

A microhabitat can strongly affect species’ distribution and community composition on a small scale. For example, changes in soil water and fertility with micro-topography can determine the spatial distribution of herbs and shrubs [57,58]. Depression is specifically related to suppressed development of some species [41,59,60]. Prior to the 1980s, permanent patches of snow were found in depressions where only mosses and lichens grew [34]. With global warming, some short-lived herb plants (e.g., *C. angustifolia*) can survive in depressions [61]. The six herb species belong to thermophilic hydro-mesophyte [62]. Therefore, we found that herbs typically occurred on semi-sunny slopes.

Studies have shown that the volcanic eruption 800 years ago in Changbai Mountain destroyed the vegetation completely and caused a primary succession of vegetation. After the volcanic disturbance, the process of succession could be divided into five stages: bare land, lichen-bryophyte community, herbaceous community, shrub-herbaceous community, and shrub community [63]. Our previous studies have also confirmed that vegetation succession stages differed on different slopes due to the varying degrees of volcanic disturbance. *R. chrysanthum* and *V. uliginosum* dominated in the climax stage of succession on the north-facing slope, and a large number of bare land and herbaceous plants distributed in the initial stage of succession on the east-facing slope [35].

## 4. Materials and Methods

### 4.1. Study Area

The Changbai Mountains (41°23’N–42°36’N, 126°55’E–129°E), located in Southeastern Jilin Province, China, rise as a mountainous boundary between China and North Korea and form the highest mountains in Northeastern China (2,691 m). The steep terrain, climate, and soil create a distinct vertical profile of vegetation types. From low to high elevations in the mountains, the vegetation changes from a mixed broadleaf–coniferous forest (600–1600 m a.s.l.), coniferous forest (1600–1800 m a.s.l.), and a subalpine mountain birch forest (1800–2050 m a.s.l.), to the alpine tundra (2050–2691 m a.s.l.).

The tundra of the Changbai Mountains is located on the upper parts of volcanic cones at elevations of more than 2000 m a.s.l. Tundra plants are well developed at elevations between 2000 and 2300 m a.s.l., where the earth’s surface consists mostly of alkaline trachyte weathered material and a small amount of ash, and has transformed by water to volcanic cone slopes. The thin loose soil substrate is formed slowly, and, recently, increased events of heavy rainfall have led to increased soil erosion, which leaves the soil thinner, dry, and impoverished [27].

### 4.2. Field Surveys

Field surveys were conducted by a systematically sampling procedure. In August 2014, a 100 m (along a contour line) × 1600 m (along the elevation) survey strip ranging from 2050 m to 2300 m a.s.l. was established within the tundra zone on the west side of the Changbai Mountain volcanic cones (Figure 6). Site selection of this big survey strip can represent a variety of habitats and topographic variability of the tundra zone. At intervals of 50 m along the slope within this strip, four 1 m × 1 m plots were laid out laterally and uniformly, and a total of 132 plots (i.e., 33 rows × 4 plots/row were established (Figure 6). The latitude and longitude, elevation (as measured by the GPS), slope, and aspect (as measured by the compass) were recorded for each plot. In each plot, the number of plant species, number of individuals, and cover (%) of each species were measured in August 2014. The plant coverage measurement used one of the most frequently-used visual estimation method.

### 4.3. Data Processing

#### 4.3.1. Determination of Dominant Species

The importance value index (IV) describes which species are the most important within the studied area, and was also determined according to the Mueller-Dombois and Ellenberg formulas (1974) [64]. Importance value = [relative density + relative frequency + relative coverage]/3, where the relative density = number of individuals of a plant/total number of plants × 100. The relative frequency = frequency of a plant population/sum of all population frequencies × 100. The relative coverage = coverage of a plant population/sum of all population coverage × 100.

#### 4.3.2. Descriptive Statistics of Dominant Species

With descriptive statistics [65], we analyzed the abundance value for the eight dominant species in each of the 132 plots, and obtained the dominant species plant distribution characteristics, including the mean value, the coefficient of variation, the skewness, and the kurtosis. Mean value describes the central tendency. The coefficient of variation describes the discrete degree. Skewness and kurtosis describe the distribution shape. The distribution and variation characteristics are indicated by the coefficient of variation (CV), skewness (SK), and kurtosis (KU). The coefficient of variation of species reflects the degree of relative variation at plot level [66]. It is generally considered that CV < 0.1 is weak variability, 0.1 ≤ CV ≤ 1.0 is moderate variability, and CV > 1.0 is strong variability [66]. We can infer the change stage or succession stage of the community by the skewness coefficient and kurtosis coefficient. The skewness of a standard normal distribution is 0. If one or more observations are extremely large, the mean of the distribution becomes larger than the median and the distribution is called positively skewed (SK > 0) [67]. If one or more observations are extremely small, the mean of the distribution becomes smaller than the median and the distribution is called negatively skewed (SK < 0) [67]. The kurtosis of a standard normal distribution is 3 [67]. It is called a leptokurtic distribution (“lepto” means slender) if KU > 3 [67]. It is called a mesokurtic distribution (“meso” means intermediate) if KU = 3 [67]. It is called a platykurtic distribution (“platy” means flat) if KU < 3 [67]. The greater the absolute value of kurtosis is, or the more extreme values of the data series are, the more deviated from the normal distribution, or the higher the probability of large fluctuations of the future this species is. When the skewness and the kurtosis of a species is close to 0 and 3, respectively, the species distribution is close to a normal distribution, which indicates a stable stage of the population or a later stage of community succession. While large skewness and kurtosis indicate an unstable phase of population (undergoing a major change) or an early period of community succession [68]. The SPSS 20.0 software was used for descriptive statistical analysis of the eight dominant species.

#### 4.3.3. The Shannon-Wiener Index (SW)

The Shannon-Wiener species diversity index (*SW*) is expressed as:(1)$$SW=-\sum_{i=1}^{S}P_{i}{\mathrm{lnP}}_{i}$$
where *S* is the number of species, *Pi* represents the diversity ratio of the *i*th species, i.e., $P_{i}=\frac{N_{i}}{N}$, where *N_i_* is the total number of individuals of the *i*th species, and *N* is the total number of individuals of all species [69].

#### 4.3.4. One-way Analysis of Variance (ANOVA) and Generalized Linear Models (GLMs)

One-way ANOVA was used to test the effects of environmental factors on the abundance of the eight dominant plant species. First, environmental factors were divided into different groups in Excel 2016 (Table 6). Then SPSS 22.0 was used to analyze the difference in abundance of plant species within each standardized environmental factor (Table 6). Multivariable linear regression was used to assess the degree of influence of different environmental factors on the distribution of different plant species by SPSS 22.0 (SPSS Inc., Chicago, IL, USA).

#### 4.3.5. Redundancy Analysis (RDA)

RDA analysis was used to analyze the relationships between plants and factors influencing their distribution [70]. In this study, abundance data related to the eight dominant species and the four environmental factors data (i.e., elevation, slope grade, slope aspect, and micro-topography) were selected for RDA analysis, using the Canoco 4.5 software.

## 5. Conclusions

The Changbai alpine shrub tundra vegetation is changing. Our results indicated that the dominance of dwarf shrubs declined, while the herbaceous species have encroached and expanded on the alpine tundra zone and have become co-dominant plant species. Herbs typically invaded and occurred on semi-sunny slopes, steep slopes, and depression. Micro-topography and slope grade had much stronger effects on herb plant distribution than elevation in the shrub tundra. Our results suggest that various micro-topographic factors associated with variations in climatic and edaphic conditions determine the spatial distribution of plants in the alpine tundra zone. Future climate warming may cause decreased snow thickness, and increased growing season length and drought stress, which may further promote replacement of the shrubs by herbs. This change could be seen as a negative effect on the Changbai alpine tundra zone in the future. Further studies need to focus on the physio-ecological mechanisms underlying the vegetation change in the alpine tundra area under global climate change.

## Figures and Tables

**Figure 1 plants-08-00370-f001:**
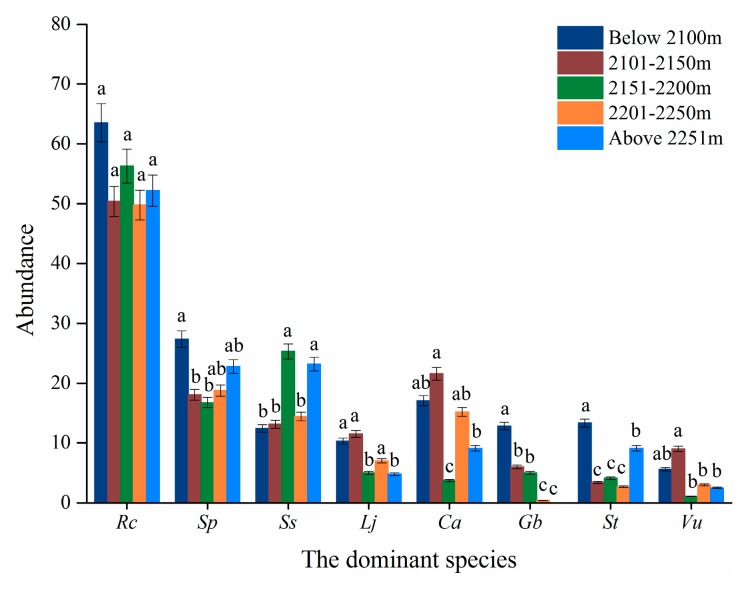
The abundance of the eight dominant species in relation to elevations. Note: Rc, indicate *R. chrysanthum*. Sp, *S. parviflora*. Ss, *S. stipulata*. Lj, *L. jamesii*. Ca, *C. angustifolia*. Gb, *G. baishanense*, St, *S. tomentosa*. Vu, *V. uliginosum*.

**Figure 2 plants-08-00370-f002:**
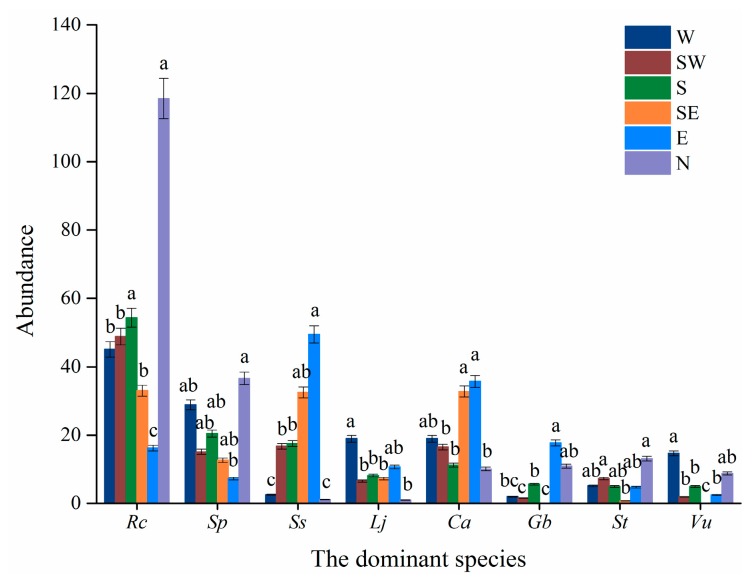
The abundance of the eight dominant species in relation to slope aspects. Note: Rc, indicate *R. chrysanthum*. Sp, *S. parviflora*. Ss, *S. stipulata*. Lj, *L. jamesii*. Ca, *C. angustifolia*. Gb, *G. baishanense*. St, *S. tomentosa*. Vu, *V. uliginosum*.

**Figure 3 plants-08-00370-f003:**
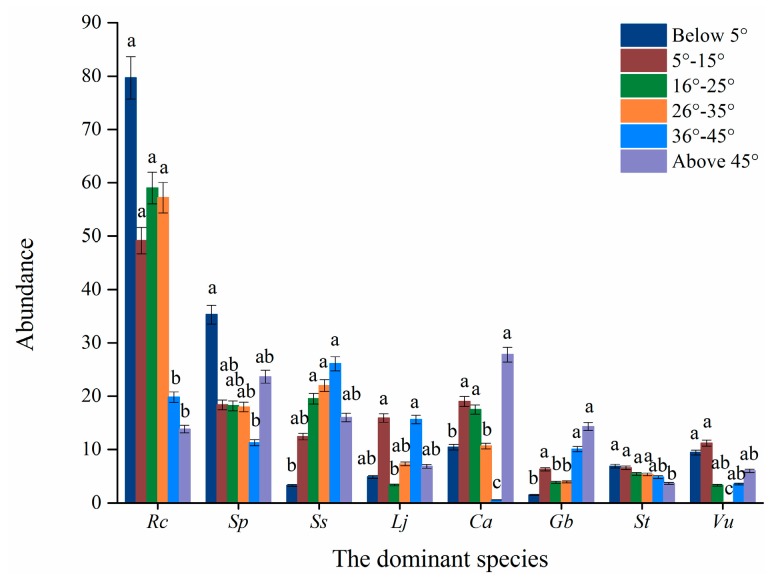
The abundance of the eight dominant species in relation to slope. Note: Rc, indicate *R. chrysanthum*. Sp, *S. parviflora*. Ss, *S. stipulata*. Lj, *L. jamesii*. Ca, *C. angustifolia*. Gb, *G. baishanense*, St, *S. tomentosa*. Vu, *V. uliginosum*.

**Figure 4 plants-08-00370-f004:**
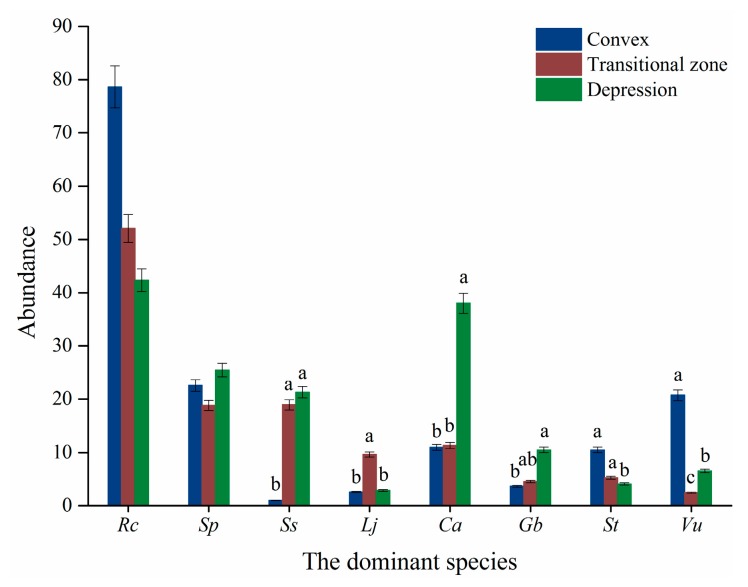
The abundance of the eight dominant species in relation to micro-topography. Note: Rc, indicate *R. chrysanthum*. Sp, *S. parviflora*. Ss, *S. stipulata*. Lj, *L. jamesii*. Ca, *C. angustifolia*. Gb, *G. baishanense*, St, *S. tomentosa*. Vu, *V. uliginosum*.

**Figure 5 plants-08-00370-f005:**
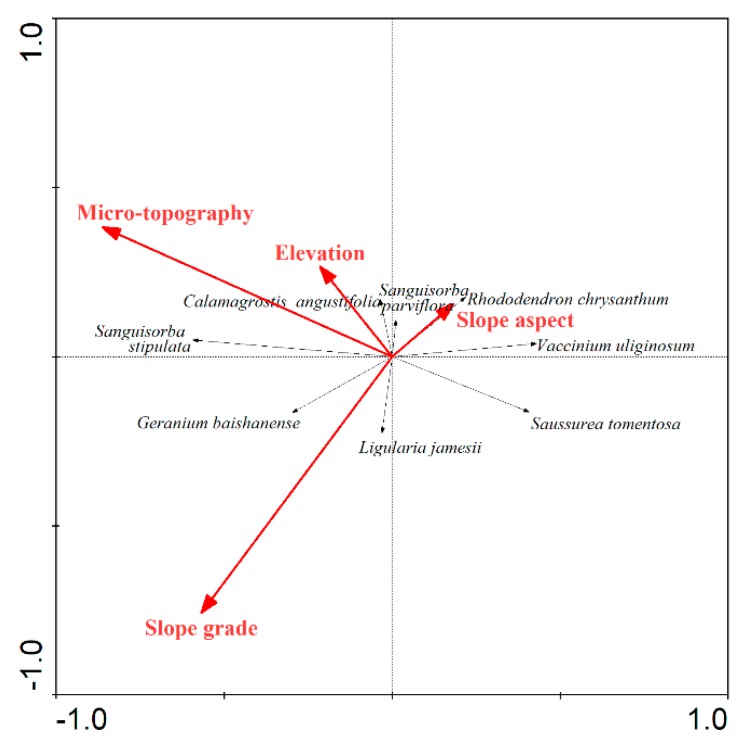
RDA ordination diagram of the eight dominant species and different environmental factors.

**Figure 6 plants-08-00370-f006:**
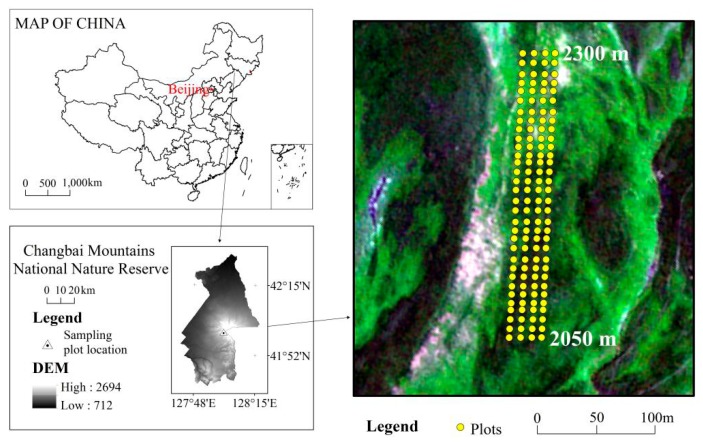
Location of the study area and sampling sites.

**Table 1 plants-08-00370-t001:** Important value index (IV) of the eight dominant species.

Plant Species	Dominant Shrubs		Dominant Herbs					
	*R. chrysanthum*	*V. uliginosum*	*S. stipulata*	*S. parviflora*	*C. angustifolia*	*L. jamesii*	*S. tomentosa*	*G. baishanense*
**IV**	20.56	3.62	8.88	8.59	6.76	4.70	3.81	3.07

**Table 2 plants-08-00370-t002:** Descriptive statistics of the abundance of the eight dominant species.

	Plant Species	Abundance (mean ± 1SE)	CV	SK	KU
**Dominant Shrubs**	*R. chrysanthum*	53.83 ± 4.67	0.95	0.53	−1.07
	*V. uliginosum*	4.81 ± 1.36	3.23	6.27	47.77
**Dominant Herbs**	*S. parviflora*	19.93 ± 1.81	1.04	1.04	0.63
	*S. stipulata*	17.28 ± 2.39	1.59	1.74	2.11
	*C. angustifolia*	14.13 ± 2.49	2.03	3.1	13.77
	*L. jamesii*	8.18 ± 1.52	2.13	3.68	17.32
	*S. tomentosa*	5.67 ± 0.83	1.69	2.22	5.55
	*G. baishanense*	5.04 ± 0.87	1.97	2.39	5.09

**Table 3 plants-08-00370-t003:** The *p* values showing the effects of environmental factors on abundance of the eight dominant plant species, tested with one-way ANOVAs.

	df	*R. chrysanthum*	*V. uliginosum*	*S. parviflora*	*S. stipulata*	*L. jamesii*	*C. angustifolia*	*S. tomentosa*	*G. baishanense*	Shrubs/Herbs	*SW*
**Elevation**	4	0.583	0.003	0.062	0.006	0.035	0.00	0.00	0.00	0.286	0.000
**Micro-toPography**	2	0.505	0.000	0.119	0.000	0.012	0.000	0.006	0.002	0.001	0.277
**Slope Aspects**	5	0.012	0.012	0.165	0.000	0.010	0.072	0.054	0.000	0.053	0.081
**Slopes**	5	0.000	0.000	0.001	0.001	0.000	0.006	0.554	0.000	0.015	0.534

**Table 4 plants-08-00370-t004:** Results of ordination based on redundancy analysis (RDA) of eight dominant species and different environmental factors, which is set in the tundra of the Changbai Mountains, Jilin, China.

Ordination Axes	Correlation Coefficient				Eigenvalues	Species-Environment Correlations	Cumulative Percentage Variance of Species-Environment Relation (%)	Monte Carlo Test	
	Elevation	Slope Aspect	Slope Grade	Micro-toPography				Test of Significance of First Canonical Axis	Test of Significance of All Canonical Axis
1	−0.143	0.06	−0.29	−0.574	0.129	0.666	75.6	*p* = 0.006	*p* = 0.002
2	0.102	0.186	−0.0383	0.146	0.021	0.382	88.1		

**Table 5 plants-08-00370-t005:** Multivariable linear regression of the eight dominant species and different environmental factors.

	Linear Regression Equation	Adjusted R Square	*Sig.*
***R. chrysanthum***	Y = 67.902 + 3.403X_1_ − 5.474X_2_ + 7.357X_3_ − 5.611X_4_	0.441	0.034
***V. uliginosum***	Y = 35.880 − 1.693X_1_ − 6.937X_2_ − 1.082X_3_ − 2.067X_4_	0.374	0.008
***S. parviflora***	Y = 13.095 + 0.699X_1_ + 5.691X_2_ + 2.323X_3_ − 2.269X_4_	0.515	0.014
***S. stipulata***	Y = −31.082 + 2.928X_1_ + 15.025X_2_ + 3.217X_3_ + 2.901X_4_	0.617	0.001
***C. angustifolia***	Y = −1.275 − 1.488X_1_ + 15.901X_2_ + 0.882X_3_ − 2.085X_4_	0.568	0.011
***L. jamesii***	Y = 28.393 − 1.955X_1_ − 0.819X_2_ − 3.146X_3_ + 0.189X_4_	0.011	0.249
***S. tomentosa***	Y = 19.978 − 0.549X_1_ − 3.932X_2_ + 0.098X_3_ − 0.696X_4_	0.45	0.033
***G. baishanense***	Y = −4.630 − 1.882X_1_ + 5.070X_2_ + 1.486X_3_ + 1.114X_4_	0.592	0.000
**Shrub/Herb ratio**	Y = 2.065 + 0.022X_1_ − 0.511X_2_ + 0.018X_3_ − 0.075X_4_	0.514	0.004
**SW**	Y = 1.192 − 0.106X_1_	0.572	0.000

**Table 6 plants-08-00370-t006:** Standardization of different environmental factors.

Elevation (m) X_1_	Micro-Topography X_2_	Aspects X_3_	Slopes (°) X_4_	Groups Assignment
2050–2100	convex	west	<5	1
2101–2150	transitional zone	southwest	6–15	2
2151–2200	depression	south	16–25	3
2201–2050		southeast	26–35	4
2051–2200		east	36–45	5
		north	>45	6

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
