# Peer review of "The Changbai Alpine Shrub Tundra Will Be Replaced by Herbaceous Tundra under Global Climate Change"

_plants, 2019, doi:10.3390/plants8100370_

Round 1

Reviewer 1 Report

NOTE: Please see the attached .docx file in case my review report here pasted would be incorrectly displayed in MDPI.
The review content is exactly the same written here

General comment and remarks

Interesting paper, pretty well written and adequate for Plants journal with data coming from sampling and not from modelling: pretty unusual nowadays when talking about climate change and, consequently, one of the main positive things of the paper. As spatial modeller I’m very glad to see that in-filed data are available and I think this kind of data might be very useful as ground-truth to build future species distribution models to forecast the (predicted) spatial distribution of plant species according to climate change scenarios.

While the paper is interesting I found the lack of any statistical test its main shortcoming. For instance while elevation, slope and aspect are acknowledged to be relevant for the distribution of the plant species (section 2.2) no statistical test were run to assess this unser a statostical point of view. I suggest to add an ANOVA or tu run a multivariate model (even linear could be fine) to test whether the influence of environmental factors (elevation, slope, etc.) is statistically significant or not (yes I guess, you have many data and you can assess it easily). PCA is fine as a descriptive analysis but it doesn’t give any statistical significance to your analysis.

Specific comments

ABSTRACT

L20-24: it is unclear which parameters the coefficient of variation, skewness etc. refer to. Please be clearer

I suggest to add the abstract why the shrubs replavcement by erbs might be positive, maybe for wildlife? Or as starting point for new forests?

INTRODUCTION

L47: shave? Maybe have?

L81: this kind of citation “Jin et al. (2016, 2018)” is not in agreement with the Journal’s style I think

RESULTS

See the general comment to improve this section. Then:

L162: how can you say “significantly” if no statistical test has been done? In other words the use of the term significantly is fine but you need to add a statistical test, even simple to discuss your data. Otherwise your paper can’t be (in my opinion) a “research article”

L236-237: this often occurs when variables has different units. In this case you need to scale your data or to use a correlation matrix with PCA instead of a classic covariance matrix (correlation matrix on raw data = covariance matrix on scaled data). Did you do this, am I right?

DISCUSSION

The section is adequate and interesting. The results are properly discussed and compared with the existing literature. The indications provided by the results are sounds but I think they should be checked in the next version of the paper once statistical test will be run to verify/certify the results.

MATERIALS AND METHODS

L352: the citation is not in agreement with the Journal’s style

L356: 1600m, wow!

L374-379: I think that all the information here provided are well known by researchers and I suggest to modify the paragraph explaining the practical meaning of such indicators in the specific study case, i.e. what does a KU>3 means for a plant species? Is it dominant? Is it gathering space? Is it going to disappear?

CONCLUSIONS

L417: “While” doesn’t need the capital letter (a simple typo mistake)

L423-425: as stated for the abstract I suggest the Authors to include a statement here specifying whether this issue could be seen as a positive or negative thing and why.

Author Response

                                                                                    September 18, 2019

Dear reviewer,

Many thanks for your comments. We have revised and improved the MS following the reviewers’ comments and carefully proofread the manuscript to minimize typographical, grammatical, and bibliographical errors. At this time, we re-submit the revised MS, and we hope to have an opportunity to publish this paper in Plants.

We have revised our manuscript according to all the comments as follows:

Reviewer 1

General comment and remarks

Interesting paper, pretty well written and adequate for Plants journal with data coming from sampling and not from modelling: pretty unusual nowadays when talking about climate change and, consequently, one of the main positive things of the paper. As spatial modeller I’m very glad to see that in-filed data are available and I think this kind of data might be very useful as ground-truth to build future species distribution models to forecast the (predicted) spatial distribution of plant species according to climate change scenarios.

While the paper is interesting I found the lack of any statistical test its main shortcoming. For instance while elevation, slope and aspect are acknowledged to be relevant for the distribution of the plant species (section 2.2) no statistical test were run to assess this unser a statostical point of view. I suggest to add an ANOVA or tu run a multivariate model (even linear could be fine) to test whether the influence of environmental factors (elevation, slope, etc.) is statistically significant or not (yes I guess, you have many data and you can assess it easily). PCA is fine as a descriptive analysis but it doesn’t give any statistical significance to your analysis.

Answer: We have revised as suggested. We added a one-way analysis of variance (ANOVA) to test whether the influence of environmental factors (elevation, slope, etc.) is statistically significant or not. Moreover, we added Generalized linear models (GLMs) to assess the degree of influence of different environmental factors on the distribution of different vegetation species. We deleted the section of PCA analysis. Also, we have rewritten this result section.

Specific comments

ABSTRACT

L20-24: it is unclear which parameters the coefficient of variation, skewness etc. refer to. Please be clearer

Answer: We have revised as suggested. “R. chrysanthum still had a relatively uniform distribution, while the herbaceous species and V. uliginosum had a patch distribution deviating from the normal distribution in the tundra zone. ”

I suggest to add the abstract why the shrubs replavcement by erbs might be positive, maybe for wildlife? Or as starting point for new forests?

Answer: We made the revisions as suggested. “Future climate warming may cause decreased snow thickness, increased growing season length and drought stress, which may further promote replacement of the shrubs by herbs, showing a retrogressive vegetation successions in the Changbai alpine tundra zone. ”

Moreover, the reason was added in discussion section 3.2.

INTRODUCTION

L47: shave? Maybe have?

Answer: We corrected.

L81: this kind of citation “Jin et al. (2016, 2018)” is not in agreement with the Journal’s style I think

Answer: We corrected. “Recently, some studies found that 8 plant species dominated the tundra, including 2 shrub species (R. chrysanthum and V. uliginosum) and 6 herb species (C. angustifolia, Geranium baishanense, Ligulariajamesii, Sanguisorba parviflora, Sanguisorbastipulata, and Saussureatomentosa) [38,39].”

RESULTS

See the general comment to improve this section. Then:

L162: how can you say “significantly” if no statistical test has been done? In other words the use of the term significantly is fine but you need to add a statistical test, even simple to discuss your data. Otherwise your paper can’t be (in my opinion) a “research article"

Answer: We appreciate the comments and have rewritten this section.We added a one-way analysis of variance (ANOVA) to test whether the influence of environmental factors (elevation, slope, etc.) is statistically significant or not.

L236-237: this often occurs when variables has different units. In this case you need to scale your data or to use a correlation matrix with PCA instead of a classic covariance matrix (correlation matrix on raw data = covariance matrix on scaled data). Did you do this, am I right?

Answer: We deleted the section of PCA analysis.

DISCUSSION

The section is adequate and interesting. The results are properly discussed and compared with the existing literature. The indications provided by the results are sounds but I think they should be checked in the next version of the paper once statistical test will be run to verify/certify the results.

Answer: We checked and made the revisions as suggested.

MATERIALS AND METHODS

L352: the citation is not in agreement with the Journal’s style

Answer: We corrected.

L356: 1600m, wow!

Answer: Yes! A100 m (along a contour line) ×1600 m (along the elevation) survey strip was established within the tundra zone on the west side of Changbai Mountain volcanic cones.

L374-379: I think that all the information here provided are well known by researchers and I suggest to modify the paragraph explaining the practical meaning of such indicators in the specific study case, i.e. what does a KU>3 means for a plant species? Is it dominant? Is it gathering space? Is it going to disappear?

Answer: We have altered the text to clarify this.As follows, “The kurtosis of a standard normal distribution is 3 [69]. It is called a leptokurtic distribution (“lepto” means slender) if KU> 3 [69]. It is called a mesokurtic distribution (“meso” means intermediate) if KU=3 [69]. It is called a platykurtic distribution (“platy” means flat) if KU<3 [69].  The greater the absolute value of kurtosis is, or the more extreme values of the data series are, the more deviated from the normal distribution, or the higher the probability of large fluctuations of future this species is. ”

CONCLUSIONS

L417: “While” doesn’t need the capital letter (a simple typo mistake)

Answer: We corrected.

L423-425: as stated for the abstract I suggest the Authors to include a statement here specifying whether this issue could be seen as a positive or negative thing and why.

Answer: We made the revisions as suggested. “Future climate warming may cause decreased snow thickness, increased growing season length and drought stress, which may further promote replacement of the shrubs by herbs, this change could be seen as a negative effect on the Changbai alpine tundra zone in the future.”

The reason was added in discussion section 3.2.

Studies have shown that the volcanic eruption 800 years ago in Changbai Mountain destroyed the vegetation totally and caused a primary succession of vegetation. After the volcanic disturbance, the process of succession could be divided into 5 stages: bare land, lichen-bryophyte community, herbaceous community, shrub-herbaceous community, shrub community[64]. Our previous studies have also confirmed that vegetation succession stages differed on different slopes due to the varying degrees of volcanic disturbance. R. chrysanthum and V. uliginosum dominated in the climax stage of succession on the north-facing slope, and a large number of bare land and herbaceous plants distributed in the initial stage of succession on the east-facing slope[65].

Sincerely yours,

Jiawei Xu

Reviewer 2 Report

The manuscript assesses the mechanisms underlying changes in the tundra vegetation of the Changbai Mountains, testing the hypothesis that topography rather than other environmental factors determines the herbaceous vegetation expansion in tundra of Changbai Mountains. Understanding how some micro-topographic factors associated with variations in climatic and edaphic conditions determine the spatial distribution of plants in the alpine tundra zone is relevant for predicting how the alpine tundra composition can change under global climate change and foresee its consequences. Therefore, I consider that this manuscript provides important insights for a better understanding of how the tundra vegetation will respond to global climate change.

I agree with the results that variations in climatic and edaphic conditions determine the spatial distribution of plants in the alpine tundra zone. However, on a small scale, changes in the soil properties associated with topography (slope gradient) determine the spatial distribution of herbs and shrubs in many ecosystems. For example, along topographic gradients in mine-slopes after restoration (see López-Marcos et al. 2019, AMBIO). Therefore, authors perhaps can include this consideration into the manuscript.

The manuscript is well written and the information well organized. I only have a few concerns that may help to improve some points.

Only a few suggestions and reflections:

Lines 11-113, Introduction. You say: ‘We aimed to test the hypothesis that topography rather than other environmental factors determines the herbaceous vegetation expansion in tundra of Changbai Mountains’, as in other environments on a small scale (see e.g. López-Marcos et al. 2019, AMBIO).

López-Marcos et al. 2019 found a topographic-successional gradient in the distribution of plant communities and plant species abundance along the reclaimed mine slope; some soil properties, related to soil water and soil organic matter, seem to be responsible for these differences.

Line 140, Results: I can’t see that ‘The abundance ratio of the 6 herbs decreased with increasing elevation (Table 3)’. Please check this and improve writing to became more specific.

Table 1: Authors don’t explain in methods how the importance value index (I.V.) was calculated. I think it is necessary.

Table 6: when you describe the factors, perhaps it would be better to use always the same terms. For exaple: Slope grade instead of Slope, Micro-topography instead of slope position, slope aspect instead of aspect, or explain well into the text when descreibind the table content (lines 254-261).

Figure 5. Please improve the quality and size of the text in figure 6. It is very difficult to read the text.

Lines 296-300, DISCUSSION: With regards to the term ‘a late invasion stage’ I want understand that you want to say that S. parviflora invaded the tundra earlier and thus it has been time to establish with a stable distribution. Contrary, S. tomentosa and G. baishanense were at an early invasion stage, which means that they are starting to invade the tundra. Is my understanding correct??? If not please revise writing to be sure that to say what you really want.

Line 313 , Discussion. Perhaps at this point (3.1. expansion of herbaceous plants) you could make a comment to the influence of changes in the soil properties associated with topography (slope gradient) determining the spatial distribution of herbs and shrubs in many ecosystems on a small scale. For example, along topographic gradients in mine-slopes after restoration (see López-Marcos et al. 2019, AMBIO).

Line 363, Methods: When sou say ‘number’ you refer to ‘number of individuals and cover (%) of each species were measured…’ Please rewrite.

Line 417, conclusions: ‘while’ instead of ‘While’.

Author Response

                                                                              September 18, 2019

Dear reviewer,

Many thanks for your comments. We have revised and improved the MS following the reviewers’ comments and carefully proofread the manuscript to minimize typographical, grammatical, and bibliographical errors. At this time, we re-submit the revised MS, and we hope to have an opportunity to publish this paper in Plants.

We have revised our manuscript according to all the comments as follows:

Reviewer 2

The manuscript assesses the mechanisms underlying changes in the tundra vegetation of the Changbai Mountains, testing the hypothesis that topography rather than other environmental factors determines the herbaceous vegetation expansion in tundra of Changbai Mountains. Understanding how some micro-topographic factors associated with variations in climatic and edaphic conditions determine the spatial distribution of plants in the alpine tundra zone is relevant for predicting how the alpine tundra composition can change under global climate change and foresee its consequences. Therefore, I consider that this manuscript provides important insights for a better understanding of how the tundra vegetation will respond to global climate change.

I agree with the results that variations in climatic and edaphic conditions determine the spatial distribution of plants in the alpine tundra zone. However, on a small scale, changes in the soil properties associated with topography (slope gradient) determine the spatial distribution of herbs and shrubs in many ecosystems. For example, along topographic gradients in mine-slopes after restoration (see López-Marcos et al. 2019, AMBIO). Therefore, authors perhaps can include this consideration into the manuscript.

The manuscript is well written and the information well organized. I only have a few concerns that may help to improve some points.

Only a few suggestions and reflections

Lines 11-113, Introduction. You say: ‘We aimed to test the hypothesis that topography rather than other environmental factors determines the herbaceous vegetation expansion in tundra of Changbai Mountains’, as in other environments on a small scale (see e.g. López-Marcos et al. 2019, AMBIO).

López-Marcos et al. 2019 found a topographic-successional gradient in the distribution of plant communities and plant species abundance along the reclaimed mine slope; some soil properties, related to soil water and soil organic matter, seem to be responsible for these differences

Answer: We appreciate your comments and thank you very much! We read the literature carefully you recommended. On a small scale, changes in the soil properties associated with topography (slope gradient) determine the spatial distribution of herbs and shrubs in many ecosystems. We are now doing some control experiments to illustrate the effect of changes in temperature, soil water and fertility on herbaceous invasion, and we will publish the result in the future. However, this study aimed to reveal the spatial distribution characteristics of 8 dominant species in the tundra mainly considering altitude and 3 topographic factors.

Line 140, Results: I can’t see that ‘The abundance ratio of the 6 herbs decreased with increasing elevation (Table 3)’. Please check this and improve writing to became more specific.

Answer: We deleted this sentence.

Table 1: Authors don’t explain in methods how the importance value index (I.V.) was calculated. I think it is necessary.

Answer: We have added this section in the Materials and Methods 4.3. Data processing.

(1)Determination of dominant species

The importance value index (IV), that describes which species are the most important within the studied area, was also determined according to Mueller-Dombois and Ellenberg formulas (1974) [66].Importance value = [relative density + relative frequency + relative coverage] / 3, where the relative density = number of individuals of a plant / total number of plants × 100; the relative frequency = frequency of a plant population / sum of all population frequencies × 100; the relative coverage = population coverage of a plant / sum of all population coverage × 100.

Table 6: when you describe the factors, perhaps it would be better to use always the same terms. For exaple: Slope grade instead of Slope, Micro-topography instead of slope position, slope aspect instead of aspect, or explain well into the text when descreibind the table content (lines 254-261).

Answer: We corrected.

Figure 5. Please improve the quality and size of the text in figure 6. It is very difficult to read the text.

Answer: We have redone figure 5 and figure 6 to improve the quality and size.

Lines 296-300, DISCUSSION: With regards to the term ‘a late invasion stage’ I want understand that you want to say that S. parviflorainvaded the tundra earlier and thus it has been time to establish with a stable distribution. Contrary, S. tomentosa and G. baishanense were at an early invasion stage, which means that they are starting to invade the tundra. Is my understanding correct??? If not please revise writing to be sure that to say what you really want.

Answer: Yes! We have altered the text to clarify this. As follows, “Statistical characteristics of the 6 herb species suggested that their invasion were not synchronous. S. parviflora invaded the tundra earlier and thus it has been time to establish with a stable distribution, indicating a late invasion stage, and it has become a primary dominant species among the 6 herb species. Compared to S. parviflora, the distribution of C. angustifolia reflected a mid-stage of invasion, and S. tomentosa and G. baishanensewere at an early invasion stage which are starting to invade the tundra. ”

Line 313 , Discussion. Perhaps at this point (3.1. expansion of herbaceous plants) you could make a comment to the influence of changes in the soil properties associated with topography (slope gradient) determining the spatial distribution of herbs and shrubs in many ecosystems on a small scale. For example, along topographic gradients in mine-slopes after restoration (see López-Marcos et al. 2019, AMBIO).

Answer: We appreciate the comments,we have added a comment in Discussion 3.2. Implication for Possible Causes of Changes in the Tundra Vegetation. As follows, “Microhabitat can strongly affect species distribution and community composition on a small scale, for example, changes in soil water and fertility with micro-topography can determine the spatial distribution of herbs and shrubs [58,59]. ”

We added a new reference: Understory response to overstory and soil gradients in mixed versus monospecifc Mediterranean pine forests. https://link.springer.com/article/10.1007/s10342-019-01215-0(First Online: 26 July 2019)

Line 363, Methods: When sou say ‘number’ you refer to ‘number of individuals and cover (%) of each species were measured…’ Please rewrite.

Answer: We have rewritten this sentence. In each plot, the number of plant species, number of individuals and cover (%) of each species were measured in August 2014.

Line 417, conclusions: ‘while’ instead of ‘While’.

Answer: We corrected.

Sincerely yours,

Jiawei Xu

Reviewer 3 Report

This study used plot surveys to analyze variations in the spatial distribution of dominant plants in the alpine tundra zone. This is a nice study. My comments listed below:

1. Authors must report the detailed methods used. For example,

(a) p. 363. How did you measure plant height and cover per species? How did you measure species frequency, abundance and dominance? Did you measure them per plot? Did you measure all individuals per plot? How easy is this task for herbaceous species?

(b) p.392-399.  Authors must add a sentence about the Important value index (IV) and its calculation.

2. Authors present descriptive statistics. Why they do not run a (or more) statistical test(s), to see if there is a significant decline or increase in abundance/frequency/dominance of dominant shrubs or herbs with elevation, slope, micro-topography etc? I think this will increase the validity of their results.

3.  I think detrended correspondence analysis (DCA) is a better analysis than PCA to understand the relationship between species composition (dominant species) and elevation/slope/aspect/micro-topography.

4. In table 6, are correlation coefficients significant or not?

5. Finally, real discussion is missing. Authors just repeat some results. Authors have to rewrite the discussion section.

Author Response

                                                                              September 18, 2019

Dear reviewer,

Many thanks for your comments. We have revised and improved the MS following the reviewers’ comments and carefully proofread the manuscript to minimize typographical, grammatical, and bibliographical errors. At this time, we re-submit the revised MS, and we hope to have an opportunity to publish this paper in Plants.

We have revised our manuscript according to all the comments as follows:

Reviewer 3

This study used plot surveys to analyze variations in the spatial distribution of dominant plants in the alpine tundra zone. This is a nice study. My comments listed below:

Authors must report the detailed methods used. For example/

(a) p. 363. How did you measure plant height and cover per species? How did you measure species frequency, abundance and dominance? Did you measure them per plot? Did you measure all individuals per plot? How easy is this task for herbaceous species?

Answer: As the reviewer said, it is not easy to conduct this large-scale field survey. We put 12 people to measure 132 plots in 7 days. The number of species in each plot, the average plant height and maximum plant height of each species, the number of individuals and the coverage of each species were measured. In the each plot, 5-10 plant of each species were randomly selected to measure as the average plant height. But plant height was deleted in the revision (4.2. Field surveys section), due to not using this index in our paper. Plant coverage measurement used one of the most frequently-used visual estimation method. Species frequency in our paper is relative frequency, which is frequency of a plant population / sum of all population frequencies × 100. By calculating the importance value of each species, the eight species with the largest significance value are selected as the dominant species

(b) p.392-399.  Authors must add a sentence about the Important value index (IV) and its calculation.

Answer: We have added this section in the Materials and Methods 4.3. Data processing.

(1)Determination of dominant species

The importance value index (IV), that describes which species are the most important within the studied area, was also determined according to Mueller-Dombois and Ellenberg formulas (1974) [66]. Importance value = [relative density + relative frequency + relative coverage] / 3, where the relative density = number of individuals of a plant / total number of plants × 100; the relative frequency = frequency of a plant population / sum of all population frequencies × 100; the relative coverage = population coverage of a plant / sum of all population coverage × 100.

Authors present descriptive statistics. Why they do not run a (or more) statistical test(s), to see if there is a significant decline or increase in abundance/frequency/dominance of dominant shrubs or herbs with elevation, slope, micro-topography etc? I think this will increase the validity of their results.

Answer: We appreciate the comments. We have added a statistical test(s) and rewritten this section. We used a one-way analysis of variance (ANOVA) and Generalized linear models (GLMs). We deleted the section of PCA analysis. Moreover, we have rewritten this result section and 4. Materials and Methods 4.3. Data processing:

(4) One-way analysis of variance (ANOVA) and Generalized linear models (GLMs)

    One-way ANOVA was used to test the effects of environmental factors on the abundance of the 8 dominant plant species. Firstly, environmental factors were divided into different groups in Excel 2016 (Table 6). Then SPSS 22.0 was used to analyze the difference in abundance of plant species within each standardized environmental factor (Table 6). Multivariable linear regression was used to assess the degree of influence of different environmental factors on the distribution of different plant species by SPSS 22.0(SPSS Inc., Chicago, IL, USA).

I think detrended correspondence analysis (DCA) is a better analysis than PCA to understand the relationship between species composition (dominant species) and elevation/slope/aspect/micro-topography.

Answer: We appreciate the comments. We deleted the section of PCA analysis.

In table 6, are correlation coefficients significant or not?

Answer: We have added the Monte Carlo test of significance.

Finally, real discussion is missing. Authors just repeat some results. Authors have to rewrite the discussion section.

Answer: We have integrated the opinions of three reviewers on the discussion part. After careful discussion, we still maintain the current discussion form but try our best to improve this part. Please see detail in the revision.

Sincerely yours,

Jiawei Xu

Round 2

Reviewer 1 Report

Dear Authors,

thanks for your replies. Your answers were clear and adequate, solving all the problems I found in the previous version of the manuscript. Thanks for adding the statistical test and for your politeness in your reply letter

Given the quality, the content and the efforts the Authors made to improve their paper, I endorse the publication of the manuscript in the Journal

Reviewer 3 Report

Authors have dealt with all my comments successfully.